# Graph Layer Security: Encrypting Information via Common Networked Physics

**DOI:** 10.3390/s22103951

**Published:** 2022-05-23

**Authors:** Zhuangkun Wei, Liang Wang, Schyler Chengyao Sun, Bin Li, Weisi Guo

**Affiliations:** 1School of Aerospace, Transport and Manufacturing, Cranfield University, Bedford MK43 0AL, UK; zhuangkun.wei@cranfield.ac.uk (Z.W.); liang.wang.133@cranfield.ac.uk (L.W.); schyler.sun@cranfield.ac.uk (S.C.S.); 2Department of Information Engineering, Beijing University of Posts and Telecommunications, Beijing 100876, China; binli@bupt.edu.cn; 3The Alan Turing Institute, London NW1 2DB, UK

**Keywords:** cyber-physical systems, wireless security, sensor network, infrastructure health monitoring, graph signal processing

## Abstract

The proliferation of low-cost Internet of Things (IoT) devices has led to a race between wireless security and channel attacks. Traditional cryptography requires high computational power and is not suitable for low-power IoT scenarios. Whilst recently developed physical layer security (PLS) can exploit common wireless channel state information (CSI), its sensitivity to channel estimation makes them vulnerable to attacks. In this work, we exploit an alternative common physics shared between IoT transceivers: the monitored channel-irrelevant physical networked dynamics (e.g., water/oil/gas/electrical signal-flows). Leveraging this, we propose, for the first time, graph layer security (GLS), by exploiting the dependency in physical dynamics among network nodes for information encryption and decryption. A graph Fourier transform (GFT) operator is used to characterise such dependency into a graph-bandlimited subspace, which allows the generation of channel-irrelevant cipher keys by maximising the secrecy rate. We evaluate our GLS against designed active and passive attackers, using IEEE 39-Bus system. Results demonstrate that GLS is not reliant on wireless CSI, and can combat attackers that have partial networked dynamic knowledge (realistic access to full dynamic and critical nodes remains challenging). We believe this novel GLS has widespread applicability in secure health monitoring and for digital twins in adversarial radio environments.

## 1. Introduction

Mass digitisation of people and things has opened the doorway to the Internet of Things (IoT), critical in many social and industrial applications [1,2]. In particular, IoT is envisaged to deliver the vital data to inform digital twins and improve infrastructure maintenance and safety. In many cases, wireless IoT sensors that measure physical signals (e.g., gas pressure, water contamination) are buried underground [3,4]. Encrypting the critical infrastructure information is important for national security, commercial sensitivity, and anti-tampering requirements. Many current IoT wireless transmissions (e.g., LoRaWAN [5], ZigBee) are vulnerable to eavesdropping. Authentication (e.g., over-the-air activation session keys in LoRaWAN, elliptic curve Diffie–Hellman in Bluetooth) verifies the user’s identity and prevents malicious users from accessing the network. Encrypted wireless transmission protects data integrity and confidentiality [6].

### 1.1. From Public Key Cryptography to Physical Layer Security

Conventional encrypted communications employ symmetric encryption such as the advanced encryption standard (AES), which relies on a secret key shared between them beforehand. Public key cryptography (PKC) is the de facto key distribution protocol, although efficient conventional PKC schemes are complex and computationally not suitable for IoT devices with limited capability [7]. This introduces not only a computational cost challenge, but also sets the IoT devices at a disadvantage against the most powerful eavesdroppers with orders of magnitude more computational power.

Physical layer security (PLS) has been proposed in recent years as a way to overcome many of the aforementioned challenges, which can be categorised as key-less PLS [8,9] and physical layer secret key generation (PL-SKG) [10,11,12]. Key-less PLS leverages the superiority of legitimate channels over wiretap ones, and tries to maximise the secrecy rate or signal to interference plus noise ratio (SINR) by steering the variables, e.g., beamforming vectors [13], trajectory of autonomous system [14], and even the phases of a reconfigurable intelligent surface (RIS) [15]. One challenge lies in that the superiority of the legitimate channel may not be always satisfied. For example, an Eve can make the channel act as adversary to legitimate nodes by introducing an Eve-controlled RIS. In such cases, key-less PLS cannot guarantee an existence of feasible solutions for secrecy rate and SINR maximisation.

Another family in PLS is PL-SKG, which exploits the reciprocal channel randomness to generate the shared secret key between legitimate nodes, without the need for a PKC management and distribution protocol [6,16,17]. PL-SKG negates the risk of key intercept and high computational requirement of PKC schemes [18], which makes it very suitable for IoT devices. PL-SKG does, however, require the wireless channel between nodes to be reciprocal (true for most propagation environments), random (small-scale fading), and unique [19,20]. This is to ensure robust symmetric key generation and avoid brute force attempts. Although PLS has been applied to a variety of embedded wireless (e.g., autonomous vehicle [21], UAV [22,23], and molecular [24]) and wired (e.g., fibre [25]) communication systems, the shortfalls are always overlooked. First, the common features used by PL-SKG for data encryption are not detached from the communications, e.g., using channel phases or received signal strength (RSS), which may provide the chances of sophisticated eavesdroppers to decode the secret key [26,27]. Second, for both key-less and PL-SKG, they require the IoT devices to make accurate estimations of the wireless channel statistics [28]. Accurate estimation requires reasonably powerful signal processing units and also requires a reasonably high communication signal-to-noise ratio (SNR). Many embedded or underground IoT devices operate in low-communication SNR regimes, and therefore PLS is not suitable.

### 1.2. Introducing Graph Layer Security (GLS): Encryption Using Networked Physics

To overcome the PLS requirement of a high-communication SNR for accurate wireless channel estimation, we identify and exploit a different physical attribute that is common to many IoT sensors. The general idea to exploit common physics has been proposed before, such as common heartbeat in different medical IoT devices across a body. However, those devices typically do not suffer from the aforementioned low-communication SNR challenges. Here, we consider IoT devices placed in underground or embedded networked systems, such as oil/gas/water pipes, electrical networks, optical fibre networks, and other underground connected systems. In networked physical systems, a common continuity equation connects all the dynamics (e.g., Navier–Stokes for flow, nonlinear Schrodinger for optic transmission, power flow for electricity). We propose to exploit the common networked physical signals at different IoT monitoring points to encrypt the IoT device’s wireless data. Such networked physics are the time-varying networked signals under each node, and are irrelevant with the wireless communications (e.g., channel phases and RSS). For example, in a smart grid, the networked physics is the electricity flow (what we will use in our simulation part). In a water distribution network, the networked physics can be the time-varying water pressure or the concentrations of some compounds. This has the following advantages: (1) IoT sensors usually have very high precision in measuring the physical signals, and (2) it requires no specific knowledge or requirement of wireless channel or public key distribution. This novel physical driven security is distinctive from both PKC and PLS, and its security independence from the digital environment makes it more resilient against digital attacks. The main contributions of this work are summarised in the following.

(1) We propose a novel digital encryption paradigm over a physical network called graph layer security (GLS). The process is data-driven and model-agnostic. We exploit the dependency of underlying networked physical dynamics to enable encryption amongst digital transceivers, without complex computations (for key generation/management in traditional cryptography-based method, or for channel estimating techniques in PLS). In GLS, the encryption and decryption is based on an additive noise generated by the wireless channel-irrelevant physical networked dynamic on one node, and can be compensated by other nodes with dependent networked dynamics. Leveraging this idea, for any pair of network nodes as Tx and Rx, we select the relay nodes with dependent dynamics, and the information then can be encrypted, reconciliated, and finally decrypted by Tx, relay, and Rx nodes.

(2) One difficulty of the relay selection lies in the dependency analysis on the random networked dynamics. To overcome this, we draw heavily on sparsifying high-dimensional random nonlinear dynamics by the use of a data-driven graph Fourier transform (GFT) operator, which is able to transform the random dynamics into a compact graph-bandlimited subspace, given the graph smoothness of the evolved networked dynamics (or its time-difference) [29]. By doing so, we pursue the dependency analysis of the random dynamics by finding the dependent rows of the GFT-based surrogate matrix.

(3) We analyse and evaluate the performance of our proposed GLS using a IEEE 39-Bus system. Both the passive eavesdroppers and the active attackers are considered, where the former is defined as intercepting the transmitted information (encrypted) as well as hacking parts of the network nodes and their dynamics, and the latter is defined to degrade the dynamic dependency by adding jamming perturbations to the network. The simulations show two results. First, the GLS performance depends mostly on the measuring accuracy of the networked dynamics, other than the complex channel estimation techniques for key generations required by the PLS. Second, GLS can protect communication security from the eavesdroppers; the bit error rate (BER) of the legitimate (Tx,Rx) pair can approach an order of $10^{-5}$ as opposed to the passive eavesdroppers ($10^{-1}$), and can still remain an order of $10^{-3}$ in the face of active dynamic attackers. This suggests a promising prospect of the proposed GLS to secure the wireless communications using the graph layer common physics.

The rest of the paper is organised as follows. In Section 2, we introduce the network dynamic model, including a governing evolution pattern that maintains the dynamic dependency, and the perturbation to keep the randomness of the networked dynamics. In Section 3, we elaborate the GLS encryption and decryption process, as well as how to select the relays for each (Tx,Rx) pair. Additionally, we define the passive eavesdroppers and active attackers and analyse their influences on the GLS performance. The simulation results are provided in Section 4. We finally conclude the whole piece of work in Section 5.

## 2. Physical Dynamic Model for Data Encryption

In this section, we introduce the physical networked dynamic model for further encryption of wireless communications. The networked dynamics are the time-varying networked signals in an IoT system (e.g., the electricity flows of the smart grid system, or a time-varying compound concentrations in the water distribution network). Such signals are irrelevant with the wireless communication aspects (i.e., the dynamic patterns in a smart grid system or water distribution network do not depend on or affect the communication channels and environment). The networked dynamic is described by the underlying network topology and the dynamic signal flow over it. The network topology is configured by a static graph, denoted by G(N,W). Here, N={1,⋯,N} represents a set of total node indices. W of size N×N is the adjacent matrix, in which the (m,n)th element $w_{m,n}\in \left\{0,1\right\}$ reflects the existence of directed link from node *n* to *m*.

Given the topology of the network, the dynamic signal over each node evolves in accordance with its self-dynamic and the coupling interactions from its neighbouring (connected) nodes. We denote the signals for all k=1,⋯,K ($K\in N^{+}$) discrete time as a matrix X of size N×K, and the evolution function from discrete time *k* to k+1 as $F:R^{N}\to R^{N}$. Then, the networked dynamic model is expressed as
(1)$$X_{:,k+1}=F\left(X_{:,k}\right)+b_{k},$$
where $X_{:,k}$ represents the *k*th column of X.

In Equation (1), $b_{k}$ is the perturbations that account for the randomness of the networked dynamics. For example, in a smart grid system (in our case-study), $b_{k}\in R^{N}$ represents the electricity usages on *N* nodes at *k* time-index. Such usages by the users are random, and thereby will make the whole networked dynamics in Equation (1) random. Here, we specify $b_{k}$ by the compositions of the unknown injection amplitude $b_{k_{i}}$ of size N×1 with respect to a random injection time (i.e., $k_{i}$), governed by the Dirac delta function, i.e.,
(2)$$b_{k}=\sum_{k_{i}\in N^{+}}b_{k_{i}}\;\cdot \;\delta \left(k-k_{i}\right).$$
In Equation (2), both the injection amplitudes and time are unknown, and may have different distributions for different dynamic systems. For this work, we do not rely on the exact distribution of the perturbations, but assume its sparse appearance (e.g., the contaminant injection into a water distribution network, or the steep variations of the power usages in an electrical bus system are sparse). As such, the combination of the networked evolution model F(·) and the random perturbation will provide the dynamic dependency as well as the randomness to secure the dynamic irrelevant wireless communications, and is hard to guess by brute force, unless the total underlying network dynamics are hacked. We will then elaborate how to pursue information encryption and decryption using the graph layer dynamics.

## 3. GLS Encryption Using Physical Networked Dynamic

Given the modelling of Equations (1) and (2), the purpose of this work is to encrypt the wireless communications of any node pair, using the underlying physical dynamics of the graph layer. The illustration of the GLS is provided by Figure 1. Before the start, the nodes need to extract their underlying physical dynamics for further encryption and communication process. For any node *n* to act as the Tx, we encrypt the desirable information s (a time-series of length *L*) via the physical networked dynamic on node *n*, denoted as $X_{n,:}$, i.e.,
(3)$$s^{*}=s+X_{n,:},$$
where $s^{*}$ is the encrypted information to be transmitted. As aforementioned, the networked dynamic $X_{n,:}$ is the time-varying networked signals under node *n* (e.g., the electricity flows of the smart grid system, or a time-varying compound concentrations in the water distribution network).

After the information encryption using Equation (3) at Tx node, it will then be transmitted and processed via a group of selected relays and their dynamics (e.g., R1 and R2 in Figure 1b). At the final Rx node, the encrypted information will be received and decrypted. As such, for the wireless communication between any node pair, i.e., (Tx,Rx) = (n,m), the idea of encryption and encryption is leveraging the linear dependency of the underlying physical dynamics in Tx, relays, and Rx nodes, i.e.,
(4)$$X_{n,:}+\sum_{i=i_{1}}^{i_{r}}\alpha_{i}^{\left(m,n\right)}\;\cdot \;X_{i,:}+X_{m,:}=0.$$
In Equation (4), $i_{1},\cdots i_{r}$ are the selected relay nodes, and $\alpha_{i}^{\left(m,n\right)}$ is the corresponding coefficient that will be determined in advance (we will discuss this in Section 3.2). As such, $s^{*}$ can be transmitted via the selected relay nodes $i_{1},\cdots ,i_{r}$, each of which processes the received information via $\alpha_{i}\;\cdot \;X_{i,:}$, and transmits the processed information to the next relay. Finally, Rx node *m* decrypts the received information via $X_{m,:}$ and derives the decrypted information $\hat{s}$, i.e.,
(5)$$\begin{array}{cc} \hat{s} & =s^{*}+\sum_{i=i_{1}}^{i_{r}}\alpha_{i}^{\left(m,n\right)}\;\cdot \;X_{i,:}+X_{m,:}\\ \; & =s+\left(X_{n,:}+\sum_{i=i_{1}}^{i_{r}}\alpha_{i}^{\left(m,n\right)}\;\cdot \;X_{i,:}+X_{m,:}\right)\\ & =s. \end{array}$$

From Equations (3)–(5), it is seen that the essence of the proposed GLS is to use the channel-irrelevant networked dynamics on each node as the additive noise for security guarantees. Attributed to the dependencies of dynamics on different nodes, such additive noise can be compensated step-by-step by the corresponding relay nodes and the Rx node, whereby $\alpha^{\left(\mathrm{Tx},\mathrm{Rx}\right)}={\left[\alpha_{i_{1}}^{\left(\mathrm{Tx},\mathrm{Rx}\right)},\cdots ,\alpha_{i_{r}}^{\left(\mathrm{Tx},\mathrm{Rx}\right)}\right]}^{T}$ accounts for the dependency coefficients. As such, to ensure the secure communication between any (Tx,Rx)$\in N^{2}$ node pair, each node *i* should save a dependency coefficient matrix of size N×N, in which the (m,n)th element is $\alpha_{i}^{\left(m,n\right)}$ representing its contribution to compensate the underlying dynamics of (Tx,Rx) = (m,n) pair. In the following, we will elaborate an off-line data-driven computation of the dependency coefficients, which avoids the use of the real-time networked dynamic for encryption, but is able to capture the networked dynamic dependency from the simulated training data.

### 3.1. GLS Secrecy Rate

The secrecy rate of the GLS encryption is the channel capacity subtraction between legitimate (Tx,Rx) pairs and the wiretap channels. Here, we consider a simplified passive eavesdropper for secrecy rate computation, where the eavesdropping can only happen by intercepting the transmitted/received signal at Tx or Rx. Other sophisticated attackers (e.g., an active attacker) are examined in Section 3.3 and in the simulations.

Given Equations (3)–(5), the legitimate channel capacity of the (Tx,Rx) communication node pair can be expressed as:(6)$$C_{\mathrm{Tx},\mathrm{Rx}}=\log_{2}\left(1+\frac{E{\left(s\right)}^{2}}{∥X^{T}\;\cdot \;\alpha^{\left(\mathrm{Tx},\mathrm{Rx}\right)}{∥}_{F}^{2}+{∥\alpha^{\left(\mathrm{Tx},\mathrm{Rx}\right)}∥}_{2}^{2}\;\cdot \;\sigma^{2}}\right),$$
In Equation (6), E(s) accounts for the expectation of the information. $\alpha^{\left(\mathrm{Tx},\mathrm{Rx}\right)}={\left[\alpha_{1}^{\left(\mathrm{Tx},\mathrm{Rx}\right)},\cdots ,\alpha_{N}^{\left(\mathrm{Tx},\mathrm{Rx}\right)}\right]}^{T}$ is the stacked weight vector of Tx node, selected relays, and Rx node, with zeros for unselected nodes. $∥X^{T}\;\cdot \;\alpha^{\left(\mathrm{Tx},\mathrm{Rx}\right)}{∥}_{F}^{2}$ (with the Frobineus norm ${∥\;\cdot \;∥}_{F}$) therefore represents the residual energy after the process of Tx, selected relays, and Rx. $∥\alpha^{\left(\mathrm{Tx},\mathrm{Rx}\right)}{∥}_{2}^{2}\;\cdot \;\sigma^{2}$ gives the weighted variance of the physical measuring noise induced by the dynamic extraction sensors, whose sampling noise is assumed to follow the Gaussian distribution with zero mean and variance as $\sigma^{2}$.

Similarly, the channel capacity from Tx to an eavesdropper that is proximate at Tx and Rx can be formulated as follows:(7)$$C_{\mathrm{eav}}^{j}=\log_{2}\left(1+\frac{E{\left(s\right)}^{2}}{∥\alpha_{j}^{\left(Tx,Rx\right)}\;\cdot \;X_{j,:}{∥}_{2}^{2}}\right),\;\;\;j\in \left\{Tx,Rx\right\}.$$
In Equation (7), we remove the noise component to achieve an upper-bound eavesdropping channel capacity, which simplifies the following secrecy rate analysis for relay node selection and their weight computation.

With the help of the legitimate and wiretap channel capacities in Equations (6) and (7), the secrecy rate is specified as their difference:(8)$$R_{\mathrm{Tx},\mathrm{Rx}}={\left[C_{\mathrm{Tx},\mathrm{Rx}}-\max_{j\in \{Tx,Rx\}}C_{\mathrm{eav}}^{j}\right]}^{+},$$
where ${\left[x\right]}^{+}=\max\left(x,0\right)$. Then, we will elaborate how to select the relay nodes and their weights by maximising the secrecy rate $R_{\mathrm{Tx},\mathrm{Rx}}$, in the absence of the networked dynamics X.

### 3.2. Relay Selection & Weight Computation

To implement the GLS node pair encryption and communication in Equations (4) and (5), one is required to select the appropriate relays and compute their weights. We do so by maximising the secrecy rate $R_{\mathrm{Tx},\mathrm{Rx}}$ formulated in Equation (8). To be specific, for each (Tx,Rx) communication pair, we compute the optimal weight vector $\alpha^{\left(Tx,Rx\right)}\in R^{N}$ by solving the following optimisation problem:(9)$$\begin{array}{c} \max_{\alpha^{\left(Tx,Rx\right)}}[\log_{2}\left(1+\frac{E{\left(s\right)}^{2}}{∥X^{T}\;\cdot \;\alpha^{\left(\mathrm{Tx},\mathrm{Rx}\right)}{∥}_{F}^{2}+{∥\alpha^{\left(\mathrm{Tx},\mathrm{Rx}\right)}∥}_{2}^{2}\;\cdot \;\sigma^{2}}\right)\\ -\max_{j\in \{Tx,Rx\}}\log_{2}\left(1+\frac{E{\left(s\right)}^{2}}{∥\alpha_{j}^{\left(Tx,Rx\right)}\;\cdot \;X_{j,:}{∥}_{2}^{2}}\right){]}^{+}. \end{array}$$
As such, the relay selection is converted to the weight computation problem whereby the non-zero weights account for the selection of the corresponding nodes.

In Equation (9), two challenges remain. First, the physical dynamic X is unknown, which makes Equation (9) difficult to solve. Second, the objective function in Equation (9) is non-concave (for maximisation) with respect to $\alpha^{\left(Tx,Rx\right)}$, although the feasible region is convex. To address the above challenges, we use a data-driven GFT operator as a surrogate matrix for X, and then transform the non-concave maximisation problem into an approximated convex optimisation. We elaborate these two in the following.

#### 3.2.1. GFT Operator-Based Surrogate

Suppose that the physical networked dynamic matrix can be decomposed as $X=\Gamma \;\cdot \;\tilde{X}$ with rank(Γ)=rank(X). Then, if Γ is only related with the dynamic model, and is therefore invariant with different initialisation cases, such a Γ can be used as the surrogate matrix for analysing the row (node) dependency of X in Equation (9).

To implement this idea, we adopt the graph spectrum analysis and the concept of graph-bandlimitedness from [30,31,32,33,34,35]. Before we start, we give a brief introduction of the graph Fourier transform (GFT) and the graph-bandlimitedness. Given a GFT operator as $U^{-1}$ (typically an orthogonal matrix of size N×N, i.e., $U^{-1}=U^{T}$), the processes of GFT and inverse GFT are specified as follows [30,31,32,33,34,35,36]:(10)$$\begin{array}{cc} \tilde{x} & =U^{-1}\;\cdot \;x, \end{array}$$(11)$$\begin{array}{cc} x & =U\;\cdot \;\tilde{x}, \end{array}$$
where x of size N×1 is the graph signal, and $\tilde{x}$ is its graph frequency response. We call x graph R-bandlimited to the GFT operator $U^{-1}$, if only the elements of $\tilde{x}$ with rows in the set R⊂{1,⋯,N} are non-zeros.

According to the work in [29], this property holds for a vast variety of the time-varying networked dynamics in real-world systems, i.e., either the original $X_{:,k}$ or the time difference $X_{:,k}-X_{:,k-1}$ are (or can be approximately treated as) graph R-bandlimited to $U^{-1}$ for all time indices *k*. Here, we denote Y as the original graph signal or the corresponding time difference, i.e.,
(12)$$\begin{array}{c} Y=\left\{\begin{array}{cc} X\; & X_{:,k}\mathrm{are}\;\mathrm{graph-bandlimited}\\ {}[X_{:,k}- & X_{:,k-1}]\;\mathrm{with}\;\mathrm{all}\;k\\ & X_{:,k}-X_{:,k-1}\mathrm{are}\;\mathrm{graph-bandlimited} \end{array}\right. \end{array}$$
where the selection is dependent on difference scenarios. As such, by denoting Y=X or $[X_{:,k}-X_{:,k-1}]\;\mathrm{with}\;\mathrm{all}\;k$, a surrogate of Y can be assigned as $\Gamma =U_{:,R}$, if $Y_{:,k}$ for all time indices *k* are graph R-bandlimited to $U^{-1}$, i.e.,
(13)$$Y_{:,k}=U_{:,R}\;\cdot \;{\tilde{Y}}_{R,k}.$$
Here, for $Y=[X_{:,k}-X_{:,k-1}]\;\mathrm{with}\;\mathrm{all}\;k$, we need to replace the original physical graph signal X with the corresponding time difference Y in all GLS encryption and further processes, i.e., Equations (3)–(9).

In Equations (10)–(13), the GFT operator $U^{-1}$ is typically assigned as the eigenvector matrix of the graph adjacent matrix denoted as W [30,32], or the graph Laplacian matrix, computed as L=diag(W·1)−W [33,34,35]. Accordingly, the graph-bandlimited set R is truncated from {1,⋯,N} to concentrate on the low-graph-bandlimited area (e.g., to minimise $x^{T}\;\cdot \;L\;\cdot \;x$ for Laplacian matrix). The problem here lies in the difficulty in measuring the actual elements in adjacent matrix W (although with the knowledge of the existence status of each link). To address this, we exploit a data-driven method, as we notice that the columns of $U_{:,R}$ in Equation (13) are the orthogonal vectors derived from the columns in Y. As such, we use *D* groups of training networked dynamics denoted as $Y^{\left(d\right)}$, d=1,⋯,D, and the GFT operator $U^{-1}$ can be computed as
(14)$$U\;\cdot \;diag\left(\left[\lambda_{1},\cdots ,\lambda_{N}\right]\right)\;\cdot \;U^{T}=\left[Y_{:,1:L}^{\left(1\right)},\cdots ,Y_{:,1:L}^{\left(D\right)}\right]\;\cdot \;{\left[Y_{:,1:L}^{\left(1\right)},\cdots ,Y_{:,1:L}^{\left(D\right)}\right]}^{T},$$
where $\lambda_{1}>\cdots \lambda_{N}$ are the descended-ordered eigenvalues. Then, by measuring the maximal rank of all the training matrix, i.e, $r=\max_{d=1\cdots ,D}rank\left(Y_{:,1:L}^{\left(d\right)}\right)$, we assign R={1,⋯,r}, and the surrogate matrix is computed as
(15)$$\Gamma =U_{:,1:r}.$$

Here, it is noteworthy that for any discrete-time $k_{i}$ with a non-zero input perturbation $b_{k_{i}}\neq 0$, the GFT-based surrogate Γ may not be able to characterise $X_{:,k}=\Gamma \;\cdot \;{\tilde{X}}_{:,k}$ or $X_{:,k}-X_{:,k-1}=\Gamma \;\cdot \;\left({\tilde{X}}_{:,k}-{\tilde{X}}_{:,k-1}\right)$, since the random $b_{k_{i}}$ may not belong to the graph-bandlimited subspace spanned by the orthogonal columns of Γ. However, given that the injection perturbation is either sparse (e.g., the contaminant injection of the water distribution network), or with small magnitudes (e.g., the power usage changes in the electrical bus system), this will not severely affect the dynamic dependency characterised by the graph-bandlimited subspace for GLS encryption

#### 3.2.2. Weight Computation

After the derivation of the physical networked dynamic surrogate in Equation (15), we elaborate the process to approximate the non-concave maximisation problem in Equation (9) into a convex minimisation form for optimal (sub-optimal) weight computation. By replacing the unknown networked dynamic matrix X with the surrogate matrix Γ, we rewrite Equation (9) as follows:(16)$$\begin{array}{c} \max_{\alpha^{\left(Tx,Rx\right)}}[\log_{2}\left(1+\frac{E{\left(s\right)}^{2}}{∥\Gamma^{T}\;\cdot \;\alpha^{\left(\mathrm{Tx},\mathrm{Rx}\right)}{∥}_{F}^{2}+{∥\alpha^{\left(\mathrm{Tx},\mathrm{Rx}\right)}∥}_{2}^{2}\;\cdot \;\sigma^{2}}\right)\\ -\max_{j\in \{Tx,Rx\}}\log_{2}\left(1+\frac{E{\left(s\right)}^{2}}{∥\alpha_{j}^{\left(Tx,Rx\right)}\;\cdot \;\Gamma_{j,:}{∥}_{2}^{2}}\right){]}^{+}. \end{array}$$

We firstly remove the operator ${\left[\;\cdot \;\right]}^{+}$, and prove that the optimal value of Equation (16) is same as that of the following form:(17)$$\begin{array}{c} \max_{\alpha^{\left(Tx,Rx\right)}}[\log_{2}\left(1+\frac{E{\left(s\right)}^{2}}{∥\Gamma^{T}\;\cdot \;\alpha^{\left(\mathrm{Tx},\mathrm{Rx}\right)}{∥}_{F}^{2}+{∥\alpha^{\left(\mathrm{Tx},\mathrm{Rx}\right)}∥}_{2}^{2}\;\cdot \;\sigma^{2}}\right)\\ -\max_{j\in \{Tx,Rx\}}\log_{2}\left(1+\frac{E{\left(s\right)}^{2}}{∥\alpha_{j}^{\left(Tx,Rx\right)}\;\cdot \;\Gamma_{j,:}{∥}_{2}^{2}}\right)]. \end{array}$$
Suppose $L_{1}$ and $L_{2}$ are the optimal values of Equation (16) and Equation (17), respectively. It is straightforward that $L_{1}\geq L_{2}$, as Equation (16) always takes the maximal value from Equation (17) and 0. Then, denote $\alpha_{*}$ as the solution of Equation (16) corresponding to the maximal value $L_{1}>0$. The value of Equation (17) at $\alpha^{*}$ equalling $L_{1}$ suggests that the maximal value of Equation (17), i.e., $L_{2}$, is no less than $L_{1}$, i.e., $L_{2}\geq L_{1}$. This, combined with the aforementioned analysis of $L_{1}\geq L_{2}$, proves that $L_{1}=L_{2}$, which validates the replacement of original objective function in Equation (16) with that of Equation (17).

The problem then is converted to minimise the opposite of Equation (17), i.e.,
(18)$$\begin{array}{c} \min_{\alpha^{\left(Tx,Rx\right)}}[-\log_{2}\left(1+\frac{E{\left(s\right)}^{2}}{∥\Gamma^{T}\;\cdot \;\alpha^{\left(\mathrm{Tx},\mathrm{Rx}\right)}{∥}_{F}^{2}+{∥\alpha^{\left(\mathrm{Tx},\mathrm{Rx}\right)}∥}_{2}^{2}\;\cdot \;\sigma^{2}}\right)\\ +\max_{j\in \{Tx,Rx\}}\log_{2}\left(1+\frac{E{\left(s\right)}^{2}}{∥\alpha_{j}^{\left(Tx,Rx\right)}\;\cdot \;\Gamma_{j,:}{∥}_{2}^{2}}\right)]. \end{array}$$
In Equation (18), it is seen that the second term is a convex function with respect to $\alpha_{j}^{\left(\mathrm{Tx},\mathrm{Rx}\right)}$. We will then make the first term of Equation (18) convex, by introducing a slack variable β. The objective problem in Equation (18) is converted as: (19)$$\begin{array}{cc} \; & \min_{\begin{array}{c} \alpha^{\left(Tx,Rx\right)}\\ \beta \end{array}}\left[-\log_{2}\left(1+\frac{E{\left(s\right)}^{2}}{\beta }\right)+\max_{j\in \{Tx,Rx\}}\log_{2}\left(1+\frac{E{\left(s\right)}^{2}}{∥\alpha_{j}^{\left(Tx,Rx\right)}\;\cdot \;\Gamma_{j,:}{∥}_{2}^{2}}\right)\right] \end{array}$$(20)$$\begin{array}{cc} \; & \;\;\;\;\;s.t.\;∥\Gamma^{T}\;\cdot \;\alpha^{\left(\mathrm{Tx},\mathrm{Rx}\right)}{∥}_{F}^{2}+{∥\alpha^{\left(\mathrm{Tx},\mathrm{Rx}\right)}∥}_{2}^{2}\;\cdot \;\sigma^{2}-\beta \leq 0. \end{array}$$
In Equation (19), β is assigned as an upper-estimator of $∥\Gamma^{T}\;\cdot \;\alpha^{\left(\mathrm{Tx},\mathrm{Rx}\right)}{∥}_{F}^{2}+{∥\alpha^{\left(\mathrm{Tx},\mathrm{Rx}\right)}∥}_{2}^{2}\;\cdot \;\sigma^{2}$, therefore making the objective function in Equation (19) an upper-estimator of that in Equation (18). As such, the minimisation problem in Equation (18) can be converted to minimising its upper-estimator in Equation (19) with the constraint in Equation (20).

Then, it is noteworthy that $-\log_{2}\left(1+E{\left(s\right)}^{2}/\beta \right)$ is a concave function with respect to β. To transform $-\log_{2}\left(1+E{\left(s\right)}^{2}/\beta \right)$ as a convex form, we adopt the first-order Taylor expansion to represent the upper-estimator. As such, the minimisation problem in Equations (19) and (20) can be approximated by minimising its convex upper-bound with the convex constraint. Accordingly, an iterative algorithm applying the successive convex optimisation method can be designed. By assuming a given initial point as $\beta_{\mathrm{ini}}$ from the last epoch, the first-order Taylor expansions of $\log_{2}\left(1+E{\left(s\right)}^{2}/\beta \right)$ at $\beta_{\mathrm{ini}}$ are expressed respectively as:(21)$$-\log_{2}\left(1+\frac{E{\left(s\right)}^{2}}{\beta }\right)\leq -\log_{2}\left(1+\frac{E{\left(s\right)}^{2}}{\beta_{\mathrm{ini}}}\right)+\frac{E{\left(s\right)}^{2}\;\cdot \;\left(\beta -\beta_{\mathrm{ini}}\right)}{\ln2\;\cdot \;\left(\beta_{\mathrm{ini}}^{2}+E{\left(s\right)}^{2}\;\cdot \;\beta_{\mathrm{ini}}\right)},$$
With the help of Equation (21), we take it into Equations (19) and (20), and add the $l_{1}$-norm of $\alpha^{\left(Tx,Rx\right)}$ to constrain the number of selected relays. The approximated convex optimisation problem is obtained as follows:(22)$$\begin{array}{cc} \; & \min_{\begin{array}{c} \alpha^{\left(Tx,Rx\right)}\\ \beta \end{array}}\left[\frac{E{\left(s\right)}^{2}\;\cdot \;\left(\beta -\beta_{\mathrm{ini}}\right)}{\ln2\;\cdot \;\left(\beta_{\mathrm{ini}}^{2}+E{\left(s\right)}^{2}\;\cdot \;\beta_{\mathrm{ini}}\right)}+\max_{j\in \{Tx,Rx\}}\log_{2}\left(1+\frac{E{\left(s\right)}^{2}}{∥\alpha_{j}^{\left(Tx,Rx\right)}\;\cdot \;\Gamma_{j,:}{∥}_{2}^{2}}\right)+{∥\alpha^{\left(\mathrm{Tx},\mathrm{Rx}\right)}∥}_{1}\right] \end{array}$$(23)$$\begin{array}{cc} \; & \;\;\;\;\;s.t.\;∥\Gamma^{T}\;\cdot \;\alpha^{\left(\mathrm{Tx},\mathrm{Rx}\right)}{∥}_{F}^{2}+{∥\alpha^{\left(\mathrm{Tx},\mathrm{Rx}\right)}∥}_{2}^{2}\;\cdot \;\sigma^{2}-\beta \leq 0. \end{array}$$

#### 3.2.3. Overall Relay Selection Algorithm

After the elaboration of the GFT-based surrogate and the weight computation, we provide the relay selection algorithm in Algorithm 1. The input is the training data of the physical networked dynamic for GFT surrogate Γ computation. Step 1 is to compute the GFT surrogate that will be used for the following weight computation. Step 2 is the initialisation for the successive convex optimisation method of weight computation. Steps 3–7 are to pursue the successive convex optimisation, in which the weight vector $\alpha^{\left(\mathrm{Tx},\mathrm{Rx}\right)}$ is successively computed by solving the convex problem in Equations (22) and (23). Then, the output is the computed weight $\alpha^{\left(\mathrm{Tx},\mathrm{Rx}\right)}$.

It is noteworthy that the proposed relay selection algorithm is in the offline manner. To be specific, the relay is selected without any knowledge of the real networked dynamics that will be used for communication encryption, but the dynamic training data from the simulator that is able to characterise the dependencies of the physical networked dynamics. In this view, two benefits are given in the following. First, the relays and their weights can be computed and saved in advance, and do not need to be re-evaluated if the hidden governing dynamic equations in Equation (1) and the network topology are fixed. As such, the computational complexity of the relay selection algorithm is trivial, and no delay will be caused by the algorithm in the securing of wireless communications. Second, the computations of the relays and their weights do not depend on the real networked dynamics that will be used for information encryption, which suggests the communication security even if the selected relays and their weights are revealed by an active or a passive attacker. We will discuss this in the following parts.
**Algorithm 1** Offline relay selection algorithm**Input:** *D* groups of training data, i.e., Y of networked dynamics.1:Compute GFT-based dynamic surrogate using Equations (14) and (15).2:Assign k=0. Find am initial feasible solution $\alpha^{\left(\mathrm{Tx},\mathrm{Rx}\right)}\left(0\right)$ and an initial slack variable β(0). Assign $∆\left(0\right)=f(\alpha^{\left(\mathrm{Tx},\mathrm{Rx}\right)}\left(0\right))$.3:**while** 
∆(k)>ϵ
**do**4:    Assign k=k+1.5:    Update $\alpha^{\left(\mathrm{Tx},\mathrm{Rx}\right)}\left(k\right)$ and β(k) by solving convex problem in Equations (22) and (23) with initial variables $\alpha^{\left(\mathrm{Tx},\mathrm{Rx}\right)}\left(k-1\right)$ and β(k−1).6:    Compute $∆\left(k\right)=f\left(\alpha^{\left(\mathrm{Tx},\mathrm{Rx}\right)}\left(k\right)\right)-f\left(\alpha^{\left(\mathrm{Tx},\mathrm{Rx}\right)}\left(k-1\right)\right)$.7:**end while**8:Assign $\alpha^{\left(\mathrm{Tx},\mathrm{Rx}\right)}\left(k\right)$ as optimal $\alpha^{\left(\mathrm{Tx},\mathrm{Rx}\right)}$.**Output:** Return $\alpha^{\left(\mathrm{Tx},\mathrm{Rx}\right)}$.

### 3.3. Active and Passive Attackers

After the elaboration of the GLS encryption, we discuss two types of the eavesdroppers, i.e., the passive and active attackers. The essence of our proposed GLS is using the signal dependency of the underlying networked dynamics to encrypt transmitted information from eavesdropping. Therefore, here we choose attackers that can either (i) passively recover the networked dynamics so as to decrypt the information, or (ii) actively destroy the signal dependency of the networked dynamics so as to impair the encryption process. Other types of attacks (e.g., spoofing, denial of services) are not within the scope of this work.

#### 3.3.1. Passive Eavesdropper

In the context of GLS, where the physical networked dynamic is used for encryption, a passive eavesdropper is defined to intercept the transmitted information without degrading the physical networked dynamics. In this view, the eavesdropper can (1) intercept only the transmitted information, or (2) in a more sophisticated way, equip eavesdropping sensors on parts of the network nodes and try to recover the whole physical networked dynamic matrix X. The former has been discussed in Equation (9), i.e., the eavesdropper can intercept the transmitted information at Tx or Rx, and the further optimal relay selection and weight computation in Equations (22) and (23) can ensure the maximal secrecy rate. We will show the GLS performance with the eavesdropper that only intercepts the transmitted information in Section 4.2.

For the latter, the GLS encryption performance will be degraded in terms of how accurately the eavesdropper recovers the unknown physical networked dynamic X, and this depends on how many network nodes have been hacked by the eavesdropping sensors. As such, the goal for this passive eavesdropper is to recover the networked dynamics and then use the recovered dynamics to decrypt the transmitted information between legitimate nodes. According to the graph sampling theory [30,31,32,33,34,35], the prerequisites of this eavesdropper are to know the data-driven GFT surrogate, Γ, and the weight vector $\alpha^{\left(Tx,Rx\right)}$ for each (Tx,Rx) pair. Then, the eavesdropping process can be divided into the following three steps.

Step 1: Eve identifies the critical subset of network nodes S⊂{1,⋯,N} using the graph sampling theory, i.e.,
(24)$$rank\left(\Gamma_{S,:}\right)=r.$$
Here, Equation (24) is implemented via a *greedy* algorithm that minimises the condition number of $\Gamma_{S,:}$ by finding and adding row index to S, i.e., S←S∪{i}, such that $i={\mathrm{argmin}}_{j\in N\setminus S}cond\left(\Gamma_{S+\{j\},:}\right)$.

Step 2: Eve deploys sensors on the selected nodes, i.e., S, and eavesdrops the physical networked dynamics under these nodes, denoted as $X_{S,:}$. Then, Eve can reconstruct the complete networked dynamics as [30,31,32,33,34,35]:(25)$$\hat{X}=\Gamma_{:,1:r}\;\cdot \;\Gamma_{S,1:r}^{†}\;\cdot \;X_{S,:},$$
which will then be used for decrypting the intercepted information.

Step 3: Eve intercepts the communication data from Tx node, and uses its reconstructed Tx underlying signals to decrypt the transmitted information.

It is noteworthy that hacking different network nodes for physical networked dynamic recovery is a strong assumption for eavesdroppers, as they have to know the data-driven GFT surrogate Γ and the weight vector $\alpha^{\left(Tx,Rx\right)}$ for each (Tx,Rx) pair, not to mention how difficult it is to embed eavesdropping sensors on a number of network nodes without being detected. We will show the number of nodes being hacked versus the GLS performance in Section 4.2

#### 3.3.2. Active Attackers

The active attacker in GLS aims to degrade the networked dynamics and their dependencies, which will subsequently deteriorate the GLS encryption that relies on them. The method is to (1) hack some of the network nodes and (2) frequently inject harmful jamming inputs from them to the networked dynamical system. Here, different from perturbation $b_{k}$ in Equation (1), the jamming injection is more frequent and with larger amplitudes. For example, in the electric bus system, an active attacker can steeply add huge and harmful power usages at a single or multiple buses, and subsequently affect the whole networked dynamic (such as rotor speed and angle).

In this view, the GLS performance depends on whether the GFT-based surrogate matrix Γ is still holding for analysing the linearly dependency of rows (i.e., relay selection and weight computation) in X. It is straightforward that an increasing rate of jamming injection will make Γ fail to represent $X=\Gamma \tilde{X}$, as a large perturbation can be artificially designed by orthogonal vectors that do not belong to the graph-bandlimited subspace spanned by the columns of Γ. We will evaluate the GLS performance versus the active attackers with different injection rates in Section 4.3.

## 4. Simulations and Results

In the simulation section, the case study is the communications among the nodes in the smart grid IEEE 39-Bus system. IEEE 39-Bus is a very general network dynamical system which has been widely used for research such as synchronisation, stability, optimal sampling, etc. [37,38,39]. This network is composed of 10 generator buses (nodes) and 29 loaded buses. When there is electricity consumption at some of the loaded buses, the angular speed under these buses will change from the standard 2π·60 rad/s, and then the cascaded effects will make all the angular speeds under all buses changes. In the meantime, 10 generator buses are used to maintain the angular speed back to the standard value, i.e., 2π·60 rad/s. These two constitute the time-varying networked dynamics (i.e., the speed deviations) of the IEEE 39-Bus system. For our GLS work, the goal here is to use the time-varying speed deviations under buses (nodes) to encrypt the wireless data transmitted among them. Here, such networked dynamical speed deviations are irrelevant with the communications (e.g., channels and RSS). For encryption and communication process, we at first compute the relay nodes and coefficients, i.e., $\alpha^{\left(Tx,Rx\right)}$ for any two pairs of (Tx,Rx) node, using Algorithm 1 in the off-line mode. Then, information is encrypted and transmitted from Tx using Equation (3), and then passed through and processed by relay nodes and Rx using Equation (5). It is noteworthy that the relay computation result for this case study is specific, and will be recomputed for different dynamical models.

### 4.1. Experimental Setting

The IEEE 39-Bus dynamics are configured in the following. Figure 2 illustrates the network structure with N=39 nodes, in which i=30,⋯,39 are generator buses, and i=1,⋯,29 are load buses.

The physical dynamic model over the network is expressed by following differential and algebraic equations (DAEs) [37,40]: (26)$$\begin{array}{cc} \; & \frac{d\theta_{i}\left(t\right)}{dt}=∆\omega_{i}\left(t\right)\;\;\;\;\;\;\;\;\;\;\;\;\;\;\;\;\;\;\;\;\;\;\;\;\;\;\;\;\;\;\;\;\;\;\;\;\;\;\;\;\;\;\;\;\;\;\;\;\;\;\;\;\;\;\;\;\;\;\;\;\;\;\;\;\;\;\;\;\;\;\;\;\;\;\;\;\;\;\;i\in \left\{1,\cdots ,39\right\} \end{array}$$(27)$$\begin{array}{cc} \; & \frac{d∆\omega_{i}\left(t\right)}{dt}=\frac{\omega_{s}}{2H_{i}}\;\cdot \;\left(T_{i}\left(t\right)-P_{i}\left(t\right)-D\;\cdot \;∆\omega_{i}\left(t\right)\right)\;\;\;\;\;\;\;\;\;\;\;\;\;\;\;\;\;\;\;\;\;\;\;\;\;\;\;\;\;\;\;\;\;\;\;\;\;\;i\in \left\{30,\cdots ,39\right\} \end{array}$$(28)$$\begin{array}{cc} \; & T_{i}\left(t\right)=-\left(K_{P}\;\cdot \;∆\omega_{i}\left(t\right)+K_{I}\int_{0}^{t}∆\omega_{i}\left(\tau \right)d\tau \right)\;\;\;\;\;\;\;\;\;\;\;\;\;\;\;\;\;\;\;\;\;\;\;\;\;\;\;\;\;\;\;\;\;\;\;\;\;\;\;\;\;\;\;i\in \left\{30,\cdots ,39\right\} \end{array}$$(29)$$\begin{array}{cc} \; & P_{i}\left(t\right)=\sum_{j=1}^{39}G_{i,j}cos\left(\theta_{i}\left(t\right)-\theta_{j}\left(t\right)\right)+B_{i,j}sin\left(\theta_{i}\left(t\right)-\theta_{j}\left(t\right)\right)\;\;\;\;\;\;\;\;\;\;\;\;\;\;\;\;\;\;\;\;\;\;\;\;\;\;\;i\in \left\{30,\cdots ,39\right\} \end{array}$$(30)$$\begin{array}{cc} \; & PL_{i}=\sum_{j=1}^{39}G_{i,j}cos\left(\theta_{i}\left(t\right)-\theta_{j}\left(t\right)\right)+B_{i,j}sin\left(\theta_{i}\left(t\right)-\theta_{j}\left(t\right)\right)\;\;\;\;\;\;\;\;\;\;\;\;\;\;\;\;\;\;\;\;\;\;\;\;\;\;\;\;\;\;i\in \left\{1,\cdots ,29\right\}. \end{array}$$
In Equations (26)–(30), $\theta_{i}\left(t\right)$ is the phasor’s angle at *i* bus, and $∆\omega_{i}\left(t\right)$ is the phasor’s speed deviation (i.e., the difference between actual speed and the standard synchronous speed $\omega_{s}=2\pi \;\cdot \;60$ rad/s). For generator node *i* (i.e., i=30,⋯,39), $H_{i}$ is the inertia constant, $T_{i}\left(t\right)$ is the mechanical torque controlling the generator’s speed deviation (with constants $K_{I}$ and $K_{P}$), and $P_{i}\left(t\right)$ is the electric air-tap torque affected by the neighbour nodes’ angles. $G_{i,j}$ and $B_{i,j}$ are, respectively, the real and imaginary parts of the (i,j)th element of admittance matrix, which characterises the admittance of power lines between buses. The aforementioned parameters are assigned according to the work in [37]. In Equation (30), $PL_{i}$ represents the load under bus *i*, following a Gaussian distribution with empirical expectation and variance, which accounts for the dynamical input of the system that affects the networked dynamics (i.e., the angles and the speed deviation). We also emphasise here that the aforementioned DAEs are only used for physical dynamic generation, and are unknown for the encryption and communication processes (e.g., process is data-driven and model-agnostic).

With the help of the dynamic model, we select the speed deviation as the physical networked signal for GLS encryption. To be specific, each bus (node) *i* is equipped with an IoT sensor (e.g., the phasor measurement unit PMU), aiming at measuring the speed deviation from time t=0 to t=5, by sampling rate $T_{s}=10^{-3}$ s [41] (i.e., discrete-time K=5000). As such, the (i,k)th element of matrix X in Equation (1) is assigned as the measured $∆\Omega_{i}\left(t=k\;\cdot \;T_{s}\right)$, with a physical measurement noise (not to be confused with communication SNR), which is used to encrypt (for Tx), process (for relays), and decrypt (for Rx) the transmitted information. In this simulation, any pair of nodes are tested as a transmitter (Tx) and a receiver (Rx). The information is assigned as the discrete on-off keying (OOK) with length K=5000. The channels for such communications are wireless and are independent with the physical networked dynamics.

For attackers, we test the aforementioned passive and active types, respectively. In the passive mode, we assume the eavesdropper can access part of nodes (e.g., 5%, 10%, and 25%) and their time-varying electricity phases. Then, the eavesdropper will try to recover the whole networked dynamics using the steps in Section 3.3.1. In the active mode, we assume the attacker can inject the jamming perturbations into the nodes in order to interfere with the dynamic pattern (as is discussed in Section 3.3.2).

### 4.2. GLS Performance with Passive Eavesdroppers

We at first evaluate the GLS performance with passive eavesdroppers mentioned in Section 3.3.1. For this experiment, four passive eavesdroppers are considered. The first one only intercepts the transmitted information without hacking any network node for the physical networked dynamics. The second to the fourth are considered to hack 5%, 10%, and 25% network nodes and their underlying physical networked dynamics to recover the whole dynamic matrix X, via the graph sampling theory in Equation (24). Here, it is noteworthy that the hacked node set S are not randomly selected, but follow Equation (24), i.e., satisfying $rank(\Gamma_{S,:})=r$. For example, in IEEE 39-Bus system, |S|=10 (or |S|/39≈25%) selected by the graph sampling theory is S={1,2,6,10,20,21,23,25,28,33}.

In Figure 3a, we provide the average BERs of legitimate (Tx,Rx) pairs and of eavesdroppers, versus the measurement noise variance (i.e., $\sigma^{2}$) of sensors for physical networked dynamic extraction. We can firstly observe that the BER of the legitimate users becomes lower (e.g., $10^{-1}$ to $10^{-5}$) as $\sigma^{2}$ decreases (from $10^{-3}$ to $10^{-5}$). This highlights the security’s dependency on sensor accuracy (i.e., physical measuring accuracy) rather than the wireless channel estimation quality or diversity (PLS) or public key security. For sensitive noise variance (as one expects of good IoT systems), the encrypted communication channel can achieve a decryption BER of ≈$10^{-5}$, five orders of magnitude better than the eavesdroppers (that only intercept the wireless transmitted information). This indicates the encryption reliability of GLS, which can be ensured solely by a cheap but accurate physical sensor (to extract and exploit the physical networked dynamics for communication security), as opposed to the existing wireless-based encryption (PLS) that requires complex and unreliable channel estimation techniques.

Then, it is seen that with the increase of nodes being hacked by the eavesdroppers, the eavesdropping BERs decrease. To be specific, the BER of eavesdroppers hacking 0% nodes (i.e., only intercept transmitted information) is higher than that hacking 5% nodes, higher than that hacking 10%, and the one with 25% hacked nodes is the lowest. This is analysed with the help of Figure 3b, where the dynamic recovery RMSEs by eavesdroppers versus the ratios of their hacked nodes are given, and that when the number of hacked nodes is larger than 25%, the RMSE converges. This is due to the graph sampling theory in Equation (24), which indicates that the appropriate node selection satisfying |S|=r=10 can guarantee the complete dynamic recovery (r=10 here is given by the data characteristic of IEEE 39-Bus with 10 generators). As such, Figure 3b gives the reason that eavesdroppers with 10/39≈25% hacked nodes can obtain the whole underlying dynamic for encryption and therefore achieve successful decryption. However, it is noteworthy that hacking such a large number of nodes is difficult for the eavesdroppers, as they have to equip their eavesdropping sensors on every network node they are interested in without being discovered. As such, our GLS method provides a novel perspective and security performance for IoT systems, by the utilisation of underlying physical networked dynamics that are hard to extract by the eavesdroppers.

In Figure 4, we provide the statistical distribution of (Tx,Rx) pairs with respect to different BER regions, i.e., <$10^{-3}$, between $10^{-3}$ and $10^{-1}$, and >$10^{-1}$. It is seen that the ratios of (Tx,Rx) pairs belonging to the low BER regions are always larger than those from the passive eavesdroppers (except for the one hacking 25% network nodes and obtaining the full physical networked dynamics). For instance, when measurement noise variance $\sigma^{2}<10^{-4}$, the ratio of (Tx,Rx) with BER less than $10^{-3}$ approaches 1, larger than those eavesdroppers (i.e., with hacking 0%, 5%, and 10% nodes). This statistical result, combined with the previous average result in Figure 4, demonstrates the encryption robustness and reliability of the proposed GLS to secure wireless communications against potential passive eavesdroppers.

### 4.3. GLS Performance with Active Attackers

We next evaluate the GLS performance against active attackers. As is described in Section 3.3.2), the active attackers here aim to degrade the dynamic dependency among network nodes by adding artificial jamming inputs into the network. In the context of the IEEE 39-Bus system, the jamming inputs are the variations of the power usages at each bus, following the Gaussian distribution with means equalling the reference power usages at each bus, and variance as 5 per unit (larger than that of the dynamic perturbation), which will then affect the underlying dynamics (i.e., the rotor speed deviations) for information encryption and decryption. Four active attackers are considered and assigned with different levels of jamming rates, i.e., 0 s${}^{-1}$, 10 s${}^{-1}$, 100 s${}^{-1}$, 200 s${}^{-1}$, and 500 s${}^{-1}$.

In Figure 5, we provide the average BERs of the legitimate (Tx,Rx) pairs versus the measurement noise variance (i.e., $\sigma^{2}$), under different levels of jamming attacks. It is firstly seen that the BERs for all levels of jamming rates become lower, as the decrease of the measurement noise variance $\sigma^{2}$. This indicates the GLS reliability that relies on the measuring accuracy of the physical networked dynamics for information encryption and decryption. Then, it is observed that with the increase of the jamming rates, the communication performance becomes worse. For example, at the point $\sigma^{2}=3.16\times 10^{-5}$, the BER increases from an order of $10^{-5}$ to $10^{-2}$ when jamming rate increases from 0 s${}^{-1}$ to 500 s${}^{-1}$. This is because (i) the burst jamming injection degrades the linear-dependant property of the dynamics among the (Tx, relays, Rx) nodes, and (ii) with such increases of the jamming inputs, deteriorates the successful decryption rate at Rx.

In Figure 6, we provide the statistical distribution of (Tx,Rx) pairs with respect to different BER regions, i.e., <$10^{-3}$, between $10^{-3}$ and $10^{-1}$, and >$10^{-1}$), under different levels of the jamming rate from the active attackers. It is seen that, given the fixed physical measurement noise variance, with the increases of the attacker’s jamming rate, the ratio of (Tx,Rx) pairs belonging to the low BER regions decreases. For instance, when measurement noise variance has an order such as $\sigma^{2}=10^{-5}$, the ratio of (Tx,Rx) with no jamming belonging to BER $<10^{-3}$ approaches 1, larger than those affected by jamming input (i.e., with jamming rate 10 s${}^{-1}$, 100 s${}^{-1}$, 200 s${}^{-1}$ and 500 s${}^{-1}$). Nevertheless, with medium jamming rates and sensitive physical measurement noise regions, as is depicted in Figure 6 (i.e., jamming rate <=100 s${}^{-1}$ and $\sigma^{2}<=10^{-4}$), the proposed GLS can still approach the 7 overhead hard-decision FEC limit (i.e., BER $\approx 4.5\times 10^{-3}$) [42], indicating an error-free BER performance with proper FEC codes. This statistical result, combined with the previous average result in Figure 5, demonstrates the encryption robustness and reliability of the proposed GLS to secure wireless communications against potential active attackers. It is also noteworthy that the results, although from the networked dynamics of the IEEE 39-Bus system, are able to show the general applicability of our GLS scheme for securing other IoT systems with correlated underlying networked dynamics, as the encryption steps/algorithm are irrelevant with the specific IEEE 39-Bus dynamic model, i.e., Equations (26)–(30).

### 4.4. Comparison with Current PLS

We next compare our proposed GLS with current PLS. From a conceptual perspective, the idea of our GLS is to use the underlying networked dynamics to encrypt the transmitted information between Alice and Bob in the IoT scenarios. This is totally different with the current PLS, which relies on the communication channels (e.g., superiority or common feature). As such, we only provide conceptual comparisons other than experimental results, and three aspects are given in the following:

(1) PLS uses the communication channel itself for encryption/security, but our GLS leverages the networked dynamics that are totally irrelevant with the communication aspects (e.g., channels, RSS). The networked dynamics can be the electricity flows in a smart grid system (as what we used in simulations), or the water pressures in a water distribution network.

(2) As described, the usage of our GLS requires an IoT with underlying networked dynamics (e.g., communications of two nodes in one smart grid network, or communications of two water junctions in one water distribution network). This is different from PLS, whereby any two legitimate users can have encrypted communications if the small-scale fading (randomness) of their channel is enough.

(3) From attacker aspects, our GLS can be eavesdropped only if an eavesdropper can reconstruct the networked dynamics under all IoT nodes, or any attackers can destroy the GLS encryption by changing the dynamic pattern (e.g., the correlations of the dynamic signals on different nodes). This is different from the attackers considered in PLS, which try to either reconstruct the common channel feature for SKG, or weaken the superiority of legitimate channels.

## 5. Conclusions

Graph layer security (GLS) is proposed for the first time here, as a way to secure the wireless communication information via the wireless channel-irrelevant networked domain physical dynamics. Our approach is premised on the exploration of the dependencies of nonlinear physical networked dynamics among the network nodes for encryption and decryption. The advantage of this approach, as described and demonstrated, is to rely solely on the IoT sensors’ accuracy in measuring the physical dynamics X (e.g., water flow rate, contamination, gas pressure, voltage) of a networked system. Over the past few decades, we have developed cheap and accurate sensors. Therefore, encrypting the wireless information by exploiting this accuracy makes sense compared to continuously and accurately estimating the wireless environment in PLS, which remains challenging for small IoT devices.

The challenge with GLS is to develop representative GFT operators that can characterise the dynamic dependency into a feasible graph-bandlimited subspace, so that the (Tx,Rx) communication node pair can use such dependent channel-irrelevant dynamic to encrypt their transmitted information. This is especially hard for those that involve PDEs. Our prior work in sparse sensing of water distribution networks has shown that GSP can be applied successfully to Navier–Stokes PDEs in water distribution networks [4]. The generality of this data-driven approach is strong as it does not require knowledge of the underlying physical dynamic model, and, indeed, many real-world systems do not have one, or involve couplings between ODEs and PDEs (e.g., electricity grid connected to a thermo energy storage).

Then, leveraging the GFT operator, encryption and decryption schemes were designed by maximising the GLS secrecy rate. The simulation results demonstrate both the robustness and the reliability of the proposed GLS, combating both the passive eavesdroppers and the active attackers, which suggests its widespread applicability in secure wireless communications over IoT systems, especially for the challenging radio environments and computational resource scarcity scenarios where traditional cryptography and PLS are less attractive. Here, one problem lies in the relay mechanism in our GLS method, which may cause delay for information transmission and may increase the rate of interception by more sophisticated eavesdroppers. The reason lies in the exploitation of solely the linear dynamic dependency. In our future work, we will study how to extract and utilise the higher-order dynamic dependency for point-to-point wireless communications over IoT systems.

## Figures and Tables

**Figure 1 sensors-22-03951-f001:**
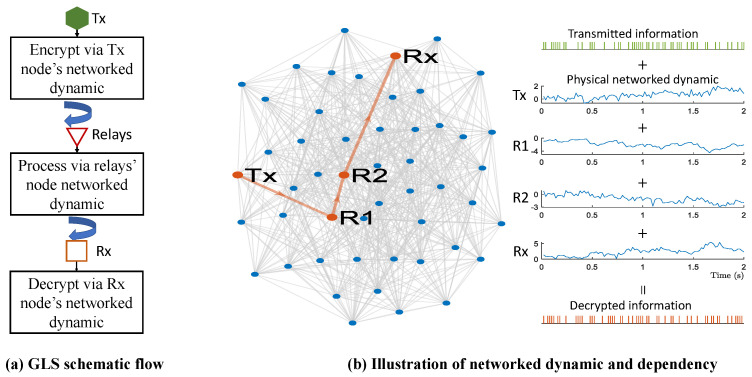
Illustration of graph layer security, where the wireless communications between (Tx,Rx) node pair is secured by the wireless channel-irrelevant physical networked dynamics. The encryption and decoding are leveraging the linear dependency of networked dynamics. (**a**) Shows the GLS schematic flow; (**b**) presents the illustration of the physical networked dynamic and the linear dependency among (Tx,Relays,Rx).

**Figure 2 sensors-22-03951-f002:**
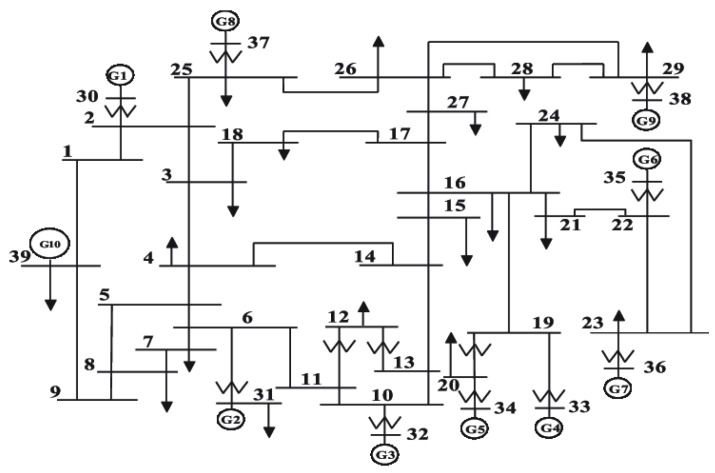
Network topology of IEEE 39-Bus system, with N=39 buses (nodes) including 10 generators (buses 30–39) and 29 load buses (1–29).

**Figure 3 sensors-22-03951-f003:**
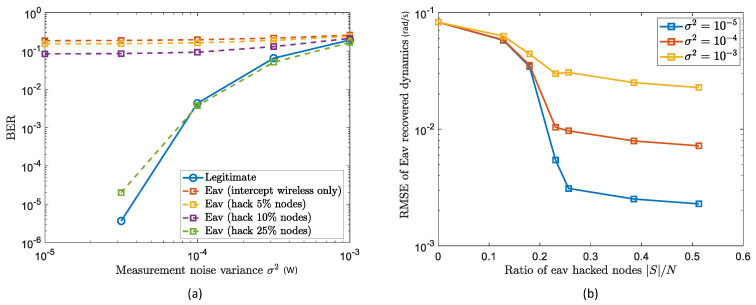
GLS performance with passive eavesdroppers: (**a**) Average BER with eavesdroppers hacking 0% nodes, 5% nodes, 10% nodes, and 25% nodes; (**b**) recovery RMSEs of physical dynamics by eavesdroppers hacking 0% nodes, 5% nodes, 10% nodes, and 25% nodes.

**Figure 4 sensors-22-03951-f004:**
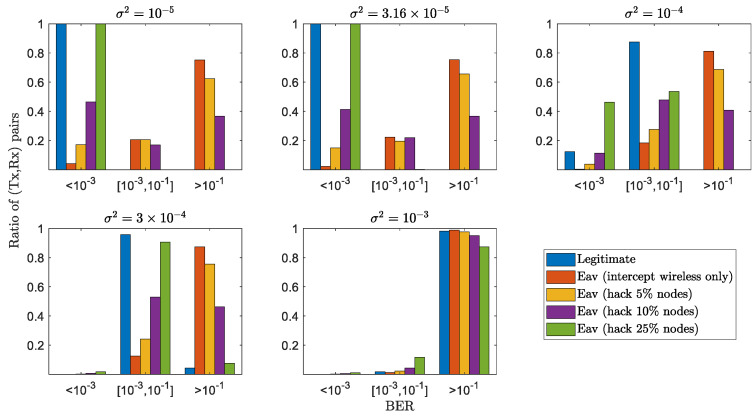
Statistical distribution of all (Tx,Rx) pairs with respect to different levels of BERs with passive Eves.

**Figure 5 sensors-22-03951-f005:**
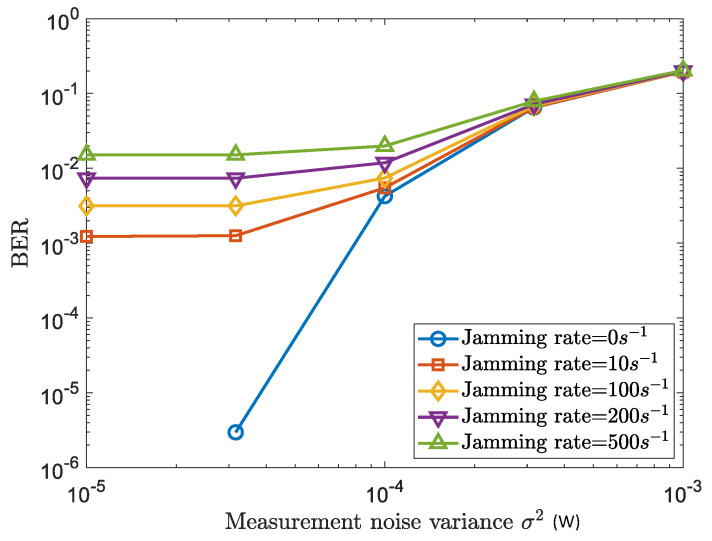
GLS performance with active attackers: Average BER versus physical measurement noise with different levels of jamming rate.

**Figure 6 sensors-22-03951-f006:**
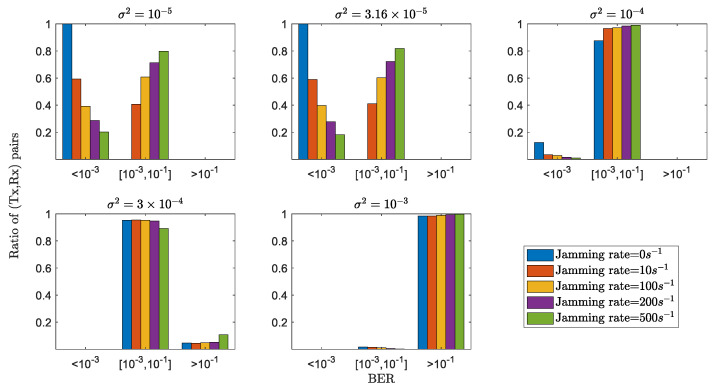
Statistical distribution of all (Tx,Rx) pairs with respect to different levels of BERs with active attackers.

## Data Availability

Data used by this study is available on request and also at: https://github.com/RookieEdward/GraphSec (accessed on 26 April 2022).

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
