# Peer review of "Graph Layer Security: Encrypting Information via Common Networked Physics"

_sensors, 2022, doi:10.3390/s22103951_

Round 1

Reviewer 1 Report

The authors address issues of physical layer security and corresponding attacks in IoT. In section 3 some description of expected passive and active attackers is presented, however it lacks more detailed and completed analytical description of possible attacks, including presentation of assumed goals, steps, resources, limitations, etc. There is also a lack of justification for choosing these attacks from the whole sphere of possible impacts.

The article considers a fairly large number of significant papers published on the field of this study. In particular, many of the referenced works have been published in the last 5 years. But some of the papers are analyzed in a too superficial way. There is also a lack of justification for the novelty of the results obtained in comparison with the analyzed references.

The experimental part is presented in a form of differential and algebraic equations with less actual details on the case study used in the work and attacks modeled on it. In Section 4, experimental data are presented quite in detail, but their analysis is not complete enough, and there is no assessment of the applicability of the results obtained.

Figure 2 is presented in a low resolution, it's difficult to recognize symbols inside the boxes and ones located near them. Besides more explanation of this figure and specific characteristics are needed.

In the diagrams in figures 2 a and b the measured quantities are shown, but they lack indication of measurement units along the vertical and horizontal axes, it should be added.

Reviewer 2 Report

  • It is not clear what the networked physics used in this paper is.
  • Whether extracting this network physics feature used in this paper is an additional process beyond normal communications?
  • It is not clear that the common continuity equation can connect all the dynamics.
  • In Eq(1), how did “b_k” come about?
  • Why is there such perturbations that account for the randomness of the dynamic.
  • The author said” we encrypt the desirable information s via the physical networked dynamic on node n”. What exactly is this physical dynamic, and is there an example to characterize it?
  • The security analysis of this physical networked dynamic is missing, i.e., whether an adversary can easily steal this feature and why the sender and receiver can get the same physical networked dynamic rather than other nodes.

Round 2

Reviewer 1 Report

In the revised paper the description of attacks has been expanded significantly, goals of the attacker as well as the steps are added. A number of new references were added into the survey of existing works, however at the most these ones are considered to superficially, one should avoid citing up to five ones simultaneously.
The comparison with already obtained results published is considerably extended and some elements of the novelty have been added into section 4.4. The experimental part has been really extended, but it still lacks of this case study details, its peculiarities, assumptions, limitations, goals, etc.

Reviewer 2 Report

My concerns have been addressed. 

Author Response

Thanks for your reviewing and your acceptance.